# The Role of the Acetylcholine System in Common Respiratory Diseases and COVID-19

**DOI:** 10.3390/molecules28031139

**Published:** 2023-01-23

**Authors:** Dehu Li, Jianghua Wu, Xianzhi Xiong

**Affiliations:** 1Department of Pulmonary and Critical Care Medicine, Union Hospital, Tongji Medical College, Huazhong University of Science and Technology, Wuhan 430022, China; 2Institute of Hematology, Union Hospital, Tongji Medical College, Huazhong University of Science and Technology, Wuhan 430022, China

**Keywords:** acetylcholine receptor, cholinergic anti-inflammatory pathway, inflammation, airway, common respiratory diseases, coronavirus disease 2019

## Abstract

As an indispensable component in human beings, the acetylcholine system regulates multiple physiological processes not only in neuronal tissues but also in nonneuronal tissues. However, since the concept of the “Nonneuronal cholinergic system (NNCS)” has been proposed, the role of the acetylcholine system in nonneuronal tissues has received increasing attention. A growing body of research shows that the acetylcholine system also participates in modulating inflammatory responses, regulating contraction and mucus secretion of respiratory tracts, and influencing the metastasis and invasion of lung cancer. In addition, the susceptibility and severity of respiratory tract infections caused by pathogens such as Mycobacterium Tuberculosis and the Severe Acute Respiratory Syndrome Coronavirus 2 (SARS-CoV-2) can also correlate with the regulation of the acetylcholine system. In this review, we summarized the major roles of the acetylcholine system in respiratory diseases. Despite existing achievements in the field of the acetylcholine system, we hope that more in-depth investigations on this topic will be conducted to unearth more possible pharmaceutical applications for the treatment of diverse respiratory diseases.

## 1. Introduction

It is well known that acetylcholine (ACh) can be released from cholinergic nerve terminals and plays an important role in the neuronal system as a neurotransmitter. As a first-discovered neurotransmitter, it regulates movements, heartbeat, digestion, and breathing, as well as other autonomic functions between neurons and non-neural tissues [1]. Based on classic physiological studies, acetylcholine can be synthesized from choline and acetyl-coenzyme A (Acetyl-CoA) by choline acetyltransferase and/or carnitine acetyltransferase (Figure 1) [2]. However, acetylcholine is rapidly degraded into choline and acetic acid since there are a large number of acetylcholine esterase (AChE) and/or butyrylcholine esterase (BChE) in interstitial spaces [3]. Meanwhile, the free choline can be reclaimed by neurons via the high-affinity choline transporter and thereafter be recycled in the process of acetylcholine biosynthesis [4]. The choline imported from the extracellular space by the high-affinity choline transporter for acetylcholine synthesis has currently been regarded as a unique source; however, it is reported that the P2X_2_ purinoceptors function as a novel pathway supplying choline for acetylcholine synthesis in the mouse retina [5]. On the other hand, calcitonin gene-related peptide (CGRP) also possesses the potential to alter acetylcholine synthesis [6]. However, ACh can also be found in the nonneuronal cholinergic system (NNCS). The concept of the “Nonneuronal cholinergic system (NNCS)” was first proposed by Sastry and Sadavongvivad in the 1970s [7]. In addition, the NNCS have been detected in reproductive organs (ovary, placenta, and amnion), immune cells, airway, and alimentary epithelial cells, as well as in cancers such as lung cancer, and breast cancer [8].

Muscarinic acetylcholine receptors (mAChRs) can be divided into five subtypes M1-M5, though the five kinds of receptors share a high degree of sequence identity in the transmembrane region, M1, M3, and M5 subtypes function with the Gq family of G proteins, and M2 and M4 subtypes work with the Gi/Go family of G proteins [9,10]. Vizi ES et al. suggested that M1 can mediate increase of acetylcholine and noradrenaline release, M2 can mediate inhibition of acetylcholine release from the Auerbach plexus and M3, which mediates inhibition of acetylcholine release from cerebral cortex and noradrenaline release from sympathetic axon terminals of the right atrium [11]. In addition, H Wang et al. provided the first molecular evidence for the presence of multiple subtypes of mAChR, including M1, M2, M3, M4 and endogenous M5 in human hearts [12]. M1 distribute in the parasympathetic ganglia, submucosal glands, and vascular endothelium in the human lungs [13,14,15]. Meanwhile, M2 mainly express pre-synaptically on parasympathetic nerve endings, where they regulate negative feedback to reduce acetylcholine release in parasympathetic ganglia and nerve terminals, besides, it can also be found in airway smooth muscle [15,16]. M3 can be detected in submucosal glands and airway smooth muscle, and M4 distribute in alveolar walls [15,17,18]. M3 are related to airway smooth muscle contraction [19,20]. In addition, M1 and M3 participate in regulating mucus secretion via vagal nerve stimulation [21]. The main effects of mAChRs in the respiratory tract are presented in Figure 2. Nicotinic acetylcholine receptors (nAChRs) are ligand-gated (cation-permeable) proteins that can be expressed in the neuronal and non-neuronal cells, including lung airway epithelial cells, macrophages, neutrophils, and muscle cells [22]. nAChRs form a variety of subtypes containing different subunit combinations. Ten alpha subunits (α1–α10), four beta subunits (β1–β4), and one delta (δ), gamma (γ), and epsilon subunit (ε) have been detected in the vertebrate species [23]. All of the subunits share a common structure including a large amino-terminal segment (about 200 residues), four transmembrane domains (less than 30 residues each), a large cytoplasmic loop (90 to 270 residues), and a variable carboxyl tail (10 to 30 residues) [24,25,26]. Among all of these receptors, α7nAChR is a homopentamer composed of five α7 monomers and it has undergone the most research [27,28]. The α7 monomer is encoded by the ten-exon CHRNA7 gene (Chr 15q13–q14), which allows the translation of a protein of about 57 kDa [29]. All mammalian α subunits but α1 have been identified in rat tracheal epithelium and the α3, α4, α7, α9, α10, β2, and β4 subunits have been detected in bronchial epithelial cells of rhesus macaques [30,31]. Expression of the α7nAChR was found in the epithelial club cells and alveolar type II cells as well as alveolar macrophages in an α7 reporter mouse strain [32]. In addition, α7nAChRs can also be found in human bronchial epithelial cells and aortic endothelial cells [33]. Niyati A Borkar et al. suggested that α7nAChR is the major nicotinic receptor expressed in human airway smooth muscle [34].

The relationship between the acetylcholine system and the respiratory system has been extensively studied in recent years. Common respiratory diseases such as chronic obstructive pulmonary disease (COPD), asthma, lung cancer, and tuberculosis have claimed countless lives of people in the past[35]. Over 5.7 million individuals have died as a result of the coronavirus disease 2019 (COVID-19) pandemic in just 24 months, primarily from respiratory causes[35]. Therefore, it requires us to constantly seek ideas to prevent and treat respiratory diseases better. In this review, we mainly elaborated on the roles of the acetylcholine system in common respiratory diseases and updated the comprehension in light of recent research.

Afferent nerve fibers transmit signals from external stimuli to the central nervous system, which then sends nerve impulses to the effector organs via the vagus nerve. Neurotransmission in the parasympathetic ganglia of the airway is mediated by acetylcholine through N (nicotinic acetylcholine receptors) and M1. Presynaptic M2 act as auto-receptors to inhibit acetylcholine release. The postganglionic neurons project primarily to mucus-producing cells and airway smooth muscle. M1 and M3 are expressed by mucus-producing cells, while M2 and M3 can be detected in airway smooth muscle. After the activation of mAChRs, the airway smooth muscle contracts, and mucus-producing cells secrete a large amount of mucus which contributes to the smaller airway diameter and limitations in lung airflow.

## 2. The Effects of Muscarinic Acetylcholine Receptors on Structural Cells and Immune Cells in the Human Airway

### 2.1. Airway Structural Cells

It has been observed that M1, M2, and M3 can be expressed in human airway epithelial cells [36]. Meanwhile, in human airway epithelial cells, the release of neutrophilic chemokines, interleukin-8 (CXCL8, IL-8), and leukotriene (LT) B_4_ can be induced by ACh through extracellular signal-regulated kinase (ERK) 1/2 or nuclear factor-kappa B (NF-κB) signal pathway and ACh can also mediate the activation of oxidative/nitrosative stress via signal transducer and activator of transcription 1 (STAT-1) pathway [37,38,39]. Moreover, M2 and M3 can be found in airway smooth muscle with an M2 predominance[40]. In human airway smooth muscle cells, muscarinic receptor agonist stimulates the release of IL-6 and CXCL8 via M3 [41]. In addition, the TGF-β1-stimulated smooth muscle proliferation can be enhanced by the activation of M2 [42]. On the other hand, the expression of M1-3 can be detected in human fibroblast cells, and the dominant expression of M2 [43]. The proliferation and collagen production in human fibroblast cells can be induced mainly via M2 stimulated with carbachol [43,44]. The effects of mAChRs on structural cells in the human airway are presented in Table 1.

### 2.2. Airway Immune Cells

Fujii et al. detected the expression of M1–M5 and nicotinic acetylcholine receptors in human lymphocytes by PCR [45]. Muscarinic receptor stimulation promotes proliferation and chemotactic activity and also regulates apoptosis [46,47]. However, the distribution of the M1-M5 on each subset of CD4+ T cells in humans has not been systemically studied. Furthermore, the expression of M1-5 has been observed through PCR, while the protein expression of M1-3 has been detected via immunostaining in human lung macrophages [48,49]. Through binding to M3, muscarinic agonist has been demonstrated to stimulate LTB_4_ release in macrophages [49,50]. On the other hand, the expression of M1-5 has been detected on neutrophils, though the expression of M1 on neutrophils still needs further confirmation [48,51]. Javier Milara et al. suggested that muscarinic agonist potentiates neutrophilic chemokines, interleukin-8 (CXCL8, IL-8) release from neutrophils [51]. In addition, it has been reported that the mRNA of M1 and M3-5 have been found in human skin mast cells and a human mast cell line [52]. However, Reinheimer et al. observed that the histamine release from mast cells can be negatively regulated by M1 in the human airway [53]. The effects of mAChRs on immune cells in the human airway are presented in Table 1.

## 3. The Anti-Inflammatory Effects of the Acetylcholine System

### 3.1. The Composition of the Cholinergic Anti-inflammatory Pathway (CAP)

Borovikova et al. first reported the cholinergic anti-inflammatory pathway exists and regulates inflammatory responses during systemic inflammation in 2000 [54]. In addition, they discovered that efferent vagus nerve signals suppress pro-inflammatory cytokine release and regarded this novel vagal function as “the cholinergic anti-inflammatory pathway” (CAP) [54]. The autonomic nervous system and the immune system communicate with each other through the CAP, which is mostly made up of parasympathetic nerves, ACh, and its receptors [55]. Previous studies showed that α7nAChR is an essential component in CAP to exert anti-inflammatory effects [56,57]. However, this still needs to be further explored to determine whether other nicotinic receptors also involve in the CAP.

### 3.2. The Basic Role of the CAP in Systemic Inflammation and Lung Inflammation

The discovery that CAP can ameliorate inflammatory responses in the body during inflammatory circumstances was based on the observations of animal experiments. Borovikova et al. found that electrical stimulation of the efferent vagus nerve suppresses the systemic inflammatory responses including the rise of both the serum and liver tumor necrosis factor (TNF)-α levels and hypotension in a rat model of septic shock induced by intravenous injection of lipopolysaccharide (LPS) [54]. Meanwhile, they found that the stimulation of the macrophages by ACh caused a concentration-dependent inhibition of the production of pro-inflammatory cytokines in vitro [54]. Wang et al. first reported that α7nAChR is essential for acetylcholine inhibition of macrophage TNF release through the CAP [56]. They observed that electrical stimulation of the vagus nerve inhibits TNF synthesis in wild-type mice but fails to inhibit TNF synthesis in alpha7-deficient mice in an LPS-induced rat septic model [56].

As for the detailed mechanism of CAP in systemic inflammation, several investigations have proved that the spleen is a key organ participating in the CAP mediated by the vagus nerve in inhibiting the production of pro-inflammatory cytokines [58,59,60]. To answer the question that how the ACh which exists in the spleen can be produced, Rosas-Ballina et al. and Vida G et al. revealed that there is a neuroimmune axis in the production of ACh and inflammatory cytokines [61,62]. Their investigations suggested that cholinergic T cells receive efferent vagus nerve signals through the norepinephrine which is released from the splenic sympathetic nerve [61,62]. Then, the norepinephrine binds to beta2-adrenoreceptors in those T cells which can promote the production of Ach [61,62]. Finally, the elevated ACh in the spleen activates α7nAChRs on macrophages, resulting in the reduction of TNF-α production [61,62].

In the lung, α7nAChRs on infiltrated inflammatory cells, such as macrophages and neutrophils can be activated by the increased acetylcholine, resulting in the amelioration of the inflammation and injury [63]. It has been also observed that activation of α7nAChR inhibited the polarization of alveolar macrophages to the pro-inflammatory M1 phenotype, whereas M2 macrophages were increased in an LPS-induced lung injury model [64,65]. In E. coli and LPS-induced lung injury mouse models, inflammatory cells such as α7nAChR+ CD11b+ monocytes and neutrophils were recruited after vagotomy, while the activation of α7nAChR by an agonist suppressed the recruitment of these cells through the phosphorylation of serine 473 of AKT1 via α7nAChR signaling [66]. Recently, Géssica Luana Antunes et al. found that neostigmine, an AChEI, reduced the levels of pro-inflammatory cytokines and conferred airway protection against oxidative stress, through the activation of CAP in an allergic asthma model [67].

### 3.3. Molecular Mechanisms and Signal Transduction Pathways of the CAP

Underlying molecular mechanisms and signaling pathways related to the CAP have been studied over these years. During the process of inflammation, the NF-κB was activated by Toll-like receptor 4 (TLR4) through MyD88-dependent signaling pathways when countering the stimulation of LPS [68]. Subsequently, phosphorylated NF-κB/p50/p65 are translocated to the nucleus to regulate the transcription of NF-κB-related genes which contributes to the production of pro-inflammatory cytokines such as IL-1β, IL-6, IL-10, TNF-α and high mobility protein box-1 (HMGB-1) in the process of inflammation (Figure 3) [69]. However, α7nAChR can counteract inflammation through the engagement of two intracellular signaling pathways: JAK2/STAT3 and PI3K/AKT in macrophages [70,71]. It has been observed that Janus kinase 2 (JAK2) can be recruited and phosphorylated after the activation of α7nAChR [70]. Upon phosphorylation, JAK2 phosphorylates and activates the signal transducer and activator of transcription 3 (STAT3), which blocks NF-κB translocation into the nucleus and the subsequent NF-κB binding to DNA [70,72]. In addition, when STAT3 is phosphorylated, it forms a dimer that translocates into the nucleus and binds to DNA, promoting the transcription of suppressor of cytokine signaling 3 (SOCS3) [70]. Furthermore, a positive regulation of nuclear factor erythroid 2-related factor 2 (Nrf2) activity has also been observed following the activation of α7nAChR [72,73]. Through PI3K/AKT signaling, α7nAChR can activate the nuclear translocation of Nrf2, which can bind to DNA and regulate the expression of responsive antioxidant genes (Figure 3) [72]. There are also many experiments supporting the anti-inflammatory effect of α7nAChR. The activation of NF-κB can be inhibited by choline in a dose-dependent manner and GTS-21(a selective α7nAChR agonist) can restrain the activation of NF-κB by preventing the translocation of NF-κBp50/p65 into the nucleus [74,75]. In addition, Wang et al. found that nicotinic stimulation prevents translocation of the NF-κB and inhibits HMGB1 secretion through the CAP which requires the involvement of α7nAChR [76]. In a 2,4,6-trinitrobenzene sulfonic acid (TNBS) mice model of colitis, Shakeeb A Wazea et al. found that galantamine (an acetylcholinesterase inhibitor) exerts anti-inflammatory/-apoptotic effects by stimulating the peripheral α7nAChR, with the participation of the JAK2/STAT3 signaling pathway [77].

Lipopolysaccharides (LPS) from bacterium can be transferred to CD14(a membrane protein immobilized by glycosylphosphatidylinositol) which subsequently leads to formatting TLR4/MD2/LPS ternary complex. Then, the receptor complex dimerizes. Thereafter, the complex induces the activation of intracellular MyD88-dependent signaling pathways which participate in the activation of NF-κB and lead to the production of pro-inflammatory cytokines. As mentioned above, the activation of JAK2/STAT3 and PI3K/AKT signaling pathway initiates the signal transduction cascade, leading to the modulation of inflammatory processes.

In general, as an essential component of the CAP, α7nAChR inhibits the production of pro-inflammatory cytokines through the JAK2/STAT3 and PI3K/AKT signaling pathways. In addition, these signal transduction pathways open a new door for drug development targeted at the CAP, which may be a promising strategy for treating respiratory diseases in the near future.

## 4. Acetylcholine System in COPD and Asthma

COPD and asthma are two of the most common chronic lung diseases worldwide. Smoking is a well-established prime risk factor for COPD and reduces the effectiveness of treatment for both asthma and COPD [78]. Smoking asthma patients have more symptoms and exacerbations than nonsmokers and smoking increases the risk of hospitalization as a result of exacerbations [79,80]. In addition, the pathophysiology of COPD and asthma is associated with the stimulation of the parasympathetic system, resulting in contraction and mucus hypersecretion of respiratory tracts. Anticholinergics are in the treatment of these diseases [81]. α7nAChR are nicotinic receptors and activated by nicotine. The studies have showed that α7nAChR agonists can exert anti-inflammatory effects in asthma and COPD animal models [82,83,84]. It seems that α7nAChR agonists could be helpful to ameliorate these inflammatory diseases. Hoi-Hin Kwok et al. reported that smoking was associated with increased α7nAChR expression [85]. However, cigarette smoke contains over 4500 different substances and nicotine is only one type of component in cigarette smoke [86,87]. It does not mean that smoking would not be detrimental for respiratory diseases. In the following content, we will further discuss the roles of the acetylcholine system participating in airway tone, mucus secretion, and inflammation responses in COPD and asthma.

### 4.1. Acetylcholine System Participates in Airway Contraction and Mucus Secretion in COPD and Asthma

Patients with COPD or asthma often feel expiratory dyspnea caused by the contraction of small airway smooth muscles. Abundant M2 and M3 can be found in airway smooth muscle, roughly in a 4:1 ratio [88]. The activation of M2 inhibits the signaling and bronchodilator effect of β2ARs by antagonizing beta-2-adrenoceptor (β2AR) activation of adenylyl cyclase. However, M2 which express pre-synaptically on parasympathetic nerve endings can constrain bronchoconstriction through negative feedback on neuronal ACh release, when activated [89]. John T Fisher et al. found that M3-knockout mice lack both methacholine and vagally induced bronchoconstriction in vivo which further elaborates the leading role of M3 in controlling airway smooth muscle contraction [90]. Airway mucus hypersecretion also called sputum production, usually can be found in COPD and asthma patients and the production of mucus is mainly controlled by the cholinergic system [91,92]. The goblet cells are the major mucus-secreting cells in the central airways which can secret mucus consisting of electrolytes, water, and high amounts of mucin. Functional M1 and M3 are expressed on the submucosal glands, roughly in a 1:2 [93,94]. Both asthma and COPD patients are characteristic of mucus hypersecretion which contributes significantly to airflow limitation by obstructing the airways [95]. Anticholinergic therapy in COPD and asthma can prevent abnormal bronchoconstriction by blocking mAChRs. Patients usually breathe easier, and signs of dyspnea get ameliorated after anticholinergics administration. In addition, it has been suggested that triple therapy consisting of inhaled corticosteroid (ICS)/long-acting β_2_-agonist (LABA)/long-acting muscarinic receptor antagonist (LAMA) provides significant clinical benefit in patients with COPD over ICS/LABA combination [96].

### 4.2. Acetylcholine System Participates in Airway Inflammation in COPD and Asthma

Except for airway contraction and mucus hypersecretion, inflammation also plays an indispensable role in the progress of COPD and asthma. In COPD, it has been observed that T lymphocytes have increased binding to ACh compared to that in normal subjects [47]. The phytohemagglutinin-stimulated Th2 cytokine release can be suppressed by tiotropium treatment in peripheral blood mononuclear cells from asthma patients [97]. As mentioned above, Loes E M Kistemaker et al. found that M3 play a pro-inflammatory role in cigarette smoke-induced neutrophilia inflammation, but M1 and M2 may possess an anti-inflammatory effect, through their investigations on cigarette smoke-induced inflammation in mice [98]. In murine models of allergic asthma, ACh contributes to allergen-induced remodeling mainly through the M3, increasing the mass of airway smooth muscle [99]. Xu et al. suggested that M3 antagonist suppressed the generation of TNF-α triggered by LPS from alveolar macrophages through regulating the NF-κB signaling pathway [100]. Therefore, these results suggest that the muscarinic acetylcholine receptors participate in the airway inflammation in COPD and asthma, and anti-muscarinic drugs might have anti-inflammatory effects.

However, it has been suggested that acetylcholine can also exert anti-inflammatory effects in COPD and asthma. A recent investigation suggested that the α7nAChRs exert anti-inflammatory effects via activating adenylyl cyclase-6 and promoting endocytosis of TLR4 mediated by a lipid raft in a COPD mouse model and indicated that the α7nAChR agonists may have the potential to become a novel therapeutic approach for COPD and possibly other inflammatory diseases [84]. On the other hand, α7nAChR-dependent signaling seems to play a role in asthma too. In an animal asthma model, the nAChR agonist 1,1-dimethyl-4-phenylpiperazinium (DMPP) was able to ameliorate the airway hyperresponsiveness, a characteristic of asthma, and airway inflammation in mice sensitized and challenged with ovalbumin (OVA) through cutting down lymphocyte and eosinophil numbers in the bronchoalveolar lavage fluid and mononuclear cell and eosinophil numbers in tissue immune cells infiltration [82]. Instead of eosinophilia, group 2 innate lymphoid cells (ILC2) play an essential role in asthma and other allergic diseases too [101]. Yuan Fang with coworkers found that a selective α7nAChR agonist, PNU-282987, can ameliorate ILC2s activation and Alternaria Alternata (AA)-induced airway inflammation [83]. However, Peiyu Sun et al. suggested that α7nAChR activation by the agonist GTS-21 enhanced the TGF-β-induced phosphorylation of Smad2/3 and transcription of fibrogenic genes in lung fibroblasts [102]. Another study demonstrated that nicotine (an α7nAChR agonist) administration promoted collagen type I expression in wild-type mice but not in α7nAChR-deficient mice in vivo [103]. In addition, Wei Hong et al. provided evidence strongly indicating that nicotine promotes human bronchial smooth muscle cells proliferation by increasing basal calcium levels and calcium influx, which result from the α7nAChR-dependent activation of the PI3K/Akt pathway and upregulation of transient receptor potential protein 6 (TRPC6) expression [104]. Therefore, these pro-fibrotic actions as well as pro-proliferative effect of airway smooth muscle via α7nAChR may be possible adverse effects to employing α7nAChR agonists in COPD and asthma.

## 5. Acetylcholine System in Lung Cancer

### 5.1. Muscarinic Acetylcholine Receptors in Lung Cancer

Through immunoblotting and RT-PCR, muscarinic acetylcholine receptor subtypes can be detected in human lung cancer cell lines [105]. Zhao et al. revealed that the activation of mAChRs by pilocarpine can induce a series of biological processes of human non-small cell lung cancer (NSCLC), such as cell proliferation, epithelial-mesenchymal transition (EMT), tumor cell migration and invasion via the MAP kinase, and the Akt pathway [106,107,108]. COPD is an independent risk factor for NSCLC, and they are closely related diseases[109,110]. Lin et al. researched the status of the M3 in NSCLC patients suffering from COPD and found that the level of M3 expressed was negatively correlated to the survival rates of NSCLC patients [111]. Meanwhile, they found that M3 promotes the invasion and migration of tumor cells through the PI3K/AKT pathway [111]. In addition, the expression of M3 was significantly related to tumor stage, Ki67 (biomarker for proliferation) expression, tumor size, lymphatic vessel size, and lymph node metastasis [105]. J. Wu et al. observed that M3 levels were elevated in metaplasia/dysplastic tissues compared to the matched normal tissues, and M3R expression was increased in Stage II and III NSCLC relative to Stage I NSCLC [112]. Recently, Meng Nie et al. demonstrated that upregulated ACh metabolism mediated drug tolerance in drug-tolerant persister (DTP) cells partly through the activation of WNT signaling in an ACh/M3-dependent manner [113]. In addition, targeting acetylcholine signaling modulates persistent drug resistance in epidermal growth factor receptor (EGFR)-mutant lung cancer and prevents tumor recurrence [113]. On the other hand, M2 signaling also possesses the potential to promote tumor growth and accelerate EMT in NSCLC [107]. Therefore, novel preparations, targeting muscarinic acetylcholine receptors may become candidate drugs to exert anti-tumor effects soon.

### 5.2. α7 Nicotinic Acetylcholine Receptors in Lung Cancer

The role of nAChR and especially the α7nAChR on cell proliferation and lung cancer has been extensively investigated in these years. The α7nAChR can be detected in several types of human lung cancer, including small cell lung cancer (SCLC) and squamous cell lung cancer cells, lung adenocarcinoma [114,115,116]. As early as 1950, Richard Doll and A. Bradford Hill reported that smoking is an important risk factor for lung cancer [117]. It has been reported that nicotine can induce the proliferation of endothelial cells [118,119]. However, in the fact that nicotine also can induce the proliferation of a variety of small cell lung carcinoma cell lines [120]. Over these years, a large number of studies have suggested that the downstream pathways of nAChR activated by nicotine can regulate tumorigenesis, angiogenesis, apoptosis, and metastasis in cancer cells [121,122]. Kwok et al. found that the activation of α7nAChR by nicotine derivative could promote programmed death-ligand 1 (PD-L1) expression in normal lung epithelial cells through STAT3/NRF2 pathways and it partly explained the cigarette smoke-induced immune evasion in lung epithelial cells [85]. In addition, after binding to α7nAChR, nicotine can induce cell invasion, migration, and epithelial-mesenchymal transition in NSCLC with the involvement of the MEK-ERK signaling pathway [123]. According to Mucchietto et al., α7nAChR activates the AKT and MEK-ERK pathways and modulates nicotine-induced cell proliferation in the A549 cell line [124]. In addition, α7nAChR exposure to nicotine increases the release of epidermal growth factor (EGF) from human normal bronchial epithelial cells[125]. Thereafter, the EGF binds to EGFR and initiates the RAS-RAF-ERK cascade, which increases cell proliferation [125]. In a word, α7nAChR, and related signaling pathways play a significant role in lung cancer, and inhibition of them seems to be a promising strategy to combat lung cancer. However, these tumor-promoting actions via α7nAChR could be possible adverse effects to employing α7nAChR agonists in COPD and asthma.

### 5.3. α5 Nicotinic Acetylcholine Receptors in Lung Cancer

Besides the α7nAChR, α5nAChR also participates in the development of lung cancer, contributing to the EMT and tumor cell metastasis in NSCLC by regulating STAT3-Jab1/Csn5 signaling [126]. It has been reported that nicotine promotes lung adenocarcinoma cell proliferation and migration via the α5nAChR/STAT3/NLRP3 inflammasome (nucleotide-binding and oligomerization domain-like receptors family) axis [127]. Zhang et al. found that the silencing of both α5nAChR and Ly6E, a member of the lymphocyte antigen 6 families and a factor in the regulation of TGF-β1/Smad signaling, can prevent NSCLC cancer cell migration, which has been confirmed in chicken embryo chorioallantoic membrane (CAM) and mouse xenograft models, and this finding may serve as a promising therapeutic intervention for non-small cell lung cancer [128]. In NSCLC, both the expression level of the hypoxia-inducible factor 1α (HIF-1α) and the vascular endothelial growth factor (VEGF) get promoted along with the activation of α5nAChR via the ERK1/2 and PI3K/Akt signaling pathways involved in tumor cell proliferation [129]. On the other hand, α5nAChR has been shown to promote cell migration and cell invasion in the lung adenoma cell line A549 [130]. Supportively, downregulation of α5nAChR inhibits lung cancer growth, at least partially, through the suppression of cyclins [131]. Zhu et al. suggested that the expression of α5nAChR is positively associated with PD-L1 and α5nAChR can mediate immune escape of lung adenocarcinoma via STAT3/Jab1-PD-L1 signaling [132]. Furthermore, they also observed that downregulation of α5nAChR suppressed the function of CD4+CD25+FOXP3+ Tregs, increased IFN-γ secretion, and enhanced T cell-mediated killing of lung adenocarcinoma cells in primary T cell cocultured system [132].

Collectively, in order to obtain other new ideas for lung cancer treatment, further studies are still needed in the future and the investigation about the acetylcholine system in lung cancer will pave the way to develop novel molecular targets and drugs in this lethal malignancy.

## 6. Acetylcholine System in Tuberculosis

Tuberculosis (TB) is a type of infectious bacterial disease caused by Mycobacterium Tuberculosis (Mtb). Studies have already shown that nicotine impairs the efforts of macrophages to control Mtb infection [133]. Meanwhile, nicotine has the potential to increase the growth of Mtb mainly in type II pneumocytes, and inhibit the expression of antimicrobial peptides in type II pneumocytes and airway basal epithelial cells partially mediated by the α7nAChR [134]. During the periods of experimental pulmonary tuberculosis, there are high concentrations of ACh and expression of choline acetyltransferase (ChAT) in lung epithelial cells and macrophages at the early infection stage, besides, the upregulation of lung ACh was even higher and coincided with ChAT and α7nAChR subunit expression in immune cells at late progressive TB [135]. Surprisingly, in vitro studies revealed that the bacteria have the potential to produce nanomolar concentrations of ACh in liquid culture and the administration of ACh/nicotinic antagonists to Mtb cultures can promote or inhibit bacterial proliferation, respectively.[135] In a word, it suggested that the upregulation of the lung non-neuronal cholinergic system can assist both bacterial growth and immunomodulation within the lung to accelerate disease progression [135]. C. E. Valdez-Miramontes et al. observed that in lung epithelial cells and macrophages infected with Mtb, nicotine treatment inhibits the production of several cytokines and chemokines such as IL-6, IL-8, IL-10, tumor necrosis factor α, C-C chemokine ligand (CCL) 2, CCL5, and C-X-C chemokine ligand (CXCL) 9 or CXCL10 in a nAChR-dependent manner [136]. However, the role of the acetylcholine system in tuberculosis still needs further study and anticholinergic medications may become candidates for anti-tuberculosis treatment in the future.

## 7. Acetylcholine System in Coronavirus Disease 2019 (COVID-19)

The disease called coronavirus disease 2019 (COVID-19) is caused by the SARS-CoV-2 virus. Previous studies have shown that coronaviruses can initiate kinds of system infections in various animals and mainly respiratory system infections in humans, such as severe acute respiratory syndrome (SARS) and the Middle East respiratory syndrome (MERS) [137,138,139]. COVID-19 is characterized by clinical heterogeneity, with severity ranging from asymptomatic patients to life-threatening diseases [140]. Some studies have revealed that there are latent associations between COVID-19 and the acetylcholine system. In addition, it has been proposed that COVID-19 may be a disease of the cholinergic nicotinic system [141].

Scientists have observed that smokers usually have a low prevalence of hospitalization among COVID-19 patients in China and they hypothesized that nicotine may exert protective effects by activating the cholinergic anti-inflammatory pathway [141,142]. Through detecting whole-blood expression of native α7 nicotinic subunit and the duplicated isoform of the human α7 neuronal nicotinic acetylcholine receptor gene (CHRFAM7A), A Courties et al. found that the expression of CHRFAM7A in critical COVID-19 patients was lower than the control (1.35  ±  2.5 versus 3.45  ±  4.2, *p* =  0.06) and COVID-19 patients not expressing CHRFAM7A subunit showed higher level of C-reactive protein (*p* =  0.04) and more pronounced lymphopenia (*p* = 0.05 for total lymphocyte, *p* =  0.03 for % live lymphocytes) [143]. They also observed that patients with no expression of CHRFAM7A often exhibit extension pulmonary lesions (> 25%) in the CT-Scan compared to those expressing CHRFAM7A (73.3% versus 44.4% respectively, *p* =  0.09) [143]. Recent studies also suggested that patients with critical COVID-19 showed significantly higher plasma ACh levels than healthy and asymptomatic participants, indicating that acetylcholine levels may be positively correlated with disease severity [144]. In addition, nicotine can attenuate the upregulation and the excessive release of pro-inflammatory cytokines/chemokines through stimulating the cholinergic anti-inflammatory reflex, and the agonists such as nicotine or GTS-21 may provide potential therapeutic access to ameliorate the dysregulated inflammatory responses in patients with severe COVID-19 [145]. However, Cattaruzza et al. suggested that the low prevalence among hospitalized patients is partially due to many smokers being misclassified as nonsmokers and claimed that tobacco smoking upregulates angiotensin converting enzyme-2 (ACE2) in a dose-dependent manner [146]. While cigarette smoking has been considered an aggravating factor in the severity of COVID-19 infections, it has been suggested that the nicotine from tobacco could ameliorate the severity of COVID-19 infection [147]. Danial Mehranfard et al. suggested that the activation of the CAP through vagus nerve stimulation (VNS) or pharmacological activation using selective α7nAChR agonists may be promising therapeutic strategies to ameliorate severe inflammation in COVID-19 patients by inhibiting the production and release of proinflammatory cytokines from macrophages, without causing systemic effects of nicotinic cholinergic receptor stimulation [147]. On the other hand, non-invasive VNS without drugs and devices has been reported to be a potential therapy option for COVID-19 patients. A single-center randomized controlled clinical trial in patients with moderate COVID-19 pneumonia suggested that reduced breathing frequency to 6/min is helpful in lowering IL-6 levels in moderate COVID-19 pneumonia without having any noticeable side effects [148].

To our knowledge, the renin–angiotensin system (RAS) has been well known for its role in regulating arterial blood pressure. However, several studies suggested that it can also participate in tissue inflammation. Through acting on Angiotensin type 1 (AT1) receptors, Angiotensin II (Ang II) induces the generation of free radicals, activates dendritic cells, promote synthesis and release of proinflammatory and chemoattractant cytokines, increase the expression of endothelial adhesion molecules and leukocyte margination and migration in tissues [30,149]. Dormoy with coworkers hypothesized that nAChR subunits act as SARS-CoV-2 spike co-receptors [150]. They reported that nAChRs domains share significant molecular similarities with the human angiotensin converting enzyme-2 (hACE2) binding domain for the SARS-CoV-2 Spike protein via sequence alignment analysis [150]. Of course, it still needs further studies to prove this hypothesis. Changux et al. hypothesized that the risk of contracting COVID-19 may be associated with the regulatory effects of cholinergic mechanisms in the RAS [151]. A study conducted by Leung et al. suggested that the expression of ACE-2 is higher in the airways of current (but not former) smokers and those with COPD [152]. However, this may increase the risk of coronavirus infections, which enter epithelial cells via the ACE-2 receptor, and it may partially explain the increased risk of respiratory system infection in active smokers and virus-related exacerbations in COPD patients [152].

Therefore, with the severe pandemic of COVID-19, we still have a lot to do to figure out the deeper implications of the interactions between the acetylcholine system and COVID-19.

## 8. Conclusion

The acetylcholine system plays a pleiotropic role in all aspects of common respiratory diseases. Activation of muscarinic and nicotinic acetylcholine receptors regulates various aspects of the development of respiratory diseases. As a major component of the CAP, α7nAChR takes part in the regulation of the inflammatory process through activation of the JAK2/STAT3 and PI3K/AKT signaling pathway. However, in addition to anti-inflammatory effects, mAChRs in the acetylcholine system also play a pro-inflammatory role in some cases. The anti-inflammatory and pro-inflammatory functions of the acetylcholine system make it a promising drug candidate. Among the common chronic respiratory diseases, COPD and asthma account for a significant proportion. The acetylcholine system participates in the pathogenesis of both diseases, and in addition, drugs targeting acetylcholine receptors have long been used in the treatment of COPD and asthma. In addition, the acetylcholine system also plays an essential role in the development of lung cancer, which has a very high morbidity and mortality rate. Activation of the corresponding acetylcholine receptors promotes tumor proliferation and metastasis. Nowadays, there are various treatments for lung cancer, such as surgery, chemotherapy, radiotherapy, immunotherapy, targeted therapy, etc. We hypothesize that drugs targeting acetylcholine receptors may also have anti-tumor effects, but of course, more research and clinical trials are needed in the future. In addition, the activation of the acetylcholine system may promote mycobacterium tuberculosis growth and accelerate the disease progression. However, the activation of the CAP helps to ameliorate the severity of COVID-19 infection. All in all, more preclinical and clinical studies are needed to elucidate the broad effects of the acetylcholine system in respiratory diseases, and there is still much to learn about targeted agents under development for the treatment of respiratory diseases.

## Figures and Tables

**Figure 1 molecules-28-01139-f001:**
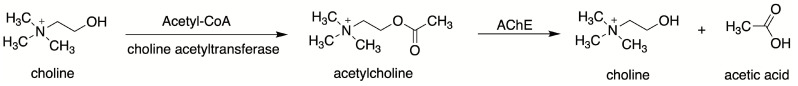
Synthesis and degradation of acetylcholine. Abbreviations: Acetyl-CoA, acetyl-coenzyme A; AChE, acetylcholine esterase.

**Figure 2 molecules-28-01139-f002:**
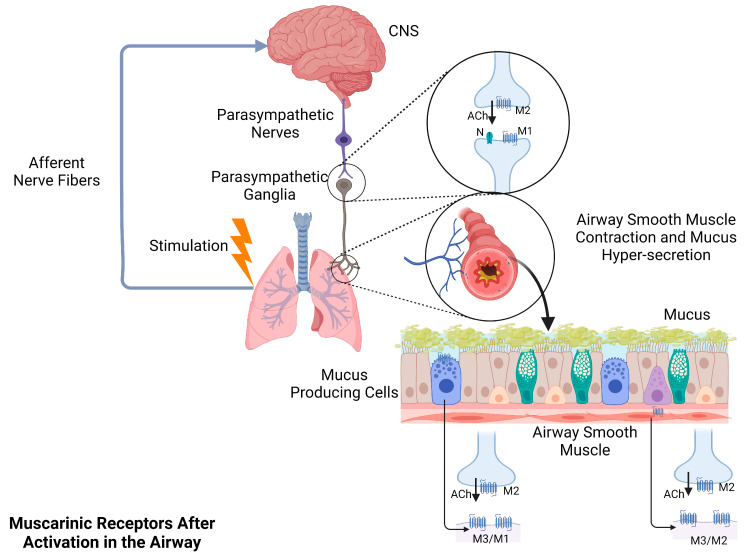
Cholinergic receptors in the regulation of airway smooth muscle contraction and mucus secretion. Abbreviations: M, muscarinic acetylcholine receptors; N, nicotinic acetylcholine receptors; ACh, acetylcholine; CNS, central nervous system. Figure created with BioRender.com (accessed on 30 December 2022).

**Figure 3 molecules-28-01139-f003:**
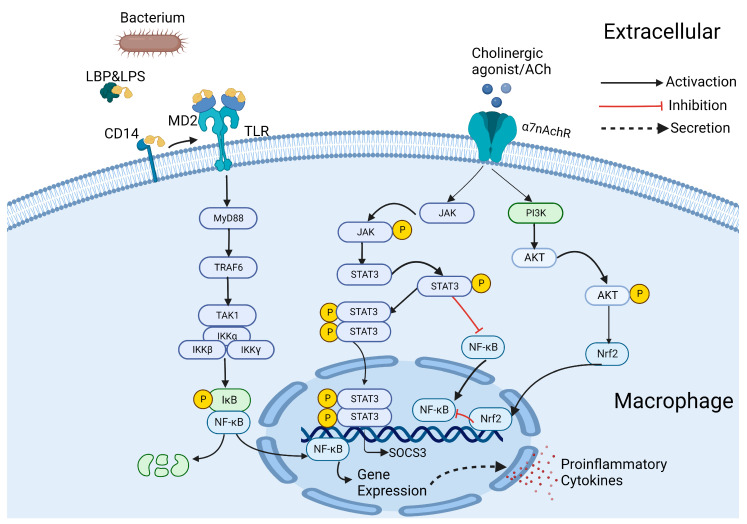
Brief Description of Signal Transduction associated with α7nAChR in the cholinergic anti-inflammatory pathway. Abbreviations: ACh, acetylcholine; LPS, lipopolysaccharide; LBP, lipopolysaccha-ride-binding protein; TLR, Toll-like receptor; MD2, myeloid differentiation protein 2; MyD88, myeloid differentiation factor 88; TRAF6, Tumor necro-sis factor receptor-associated factors 6; TAK1, TGF-beta-activated kinase-1; NF-κB, nuclear factor-κB; IκB, NF-kappa-B inhibitor; IKK, inhibitor of IκB kinases; PI3K, phosphatidylinositol-3-kinase; AKT, protein kinase B; STAT3, signal transducer and activator of transcription 3; JAK, Janus Kinase; Nrf2, nuclear factor erythroid 2–related factor 2; SOCS3, suppressor of cytokine signaling 3; P, phosphorylation. Figure created with BioRender.com (accessed on December 30th, 2022).

**Table 1 molecules-28-01139-t001:** The effects of mAChRs on structural cells and immune cells in the human airway.

Cell Type	Muscarinic Receptor Subtypes	Function	References
airway epithelial cells	M1/M2/M3	facilitation of CXCL8, LTB_4_ release activation of oxidative/nitrosative stress, mucus secretion	[21,36,37,38,39]
airway smooth muscle	M2/M3	contraction, proliferation, facilitation of IL-6 and CXCL8 release	[40,41,42]
fibroblast cells	M1/M2/M3	proliferation, collagen production	[43,44]
lymphocytes	M1/M2/M3/M4/M5	proliferation, chemotactic, apoptosis regulation	[45,46,47]
macrophages	M1/M2/M3/M4/M5	facilitation of LTB_4_ release	[48,49,50]
neutrophils	M1/M2/M3/M4/M5	facilitation of CXCL8 release	[48,51]
mast cells	M1	inhibitory regulation of histamine release	[53]

Abbreviations: M, muscarinic acetylcholine receptors; CXCL8, neutrophilic chemokines; LTB_4_, leukotriene B_4_; IL-6, interleukin-6

## Data Availability

Not applicable.

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
