# Peer review of "The Role of the Acetylcholine System in Common Respiratory Diseases and COVID-19"

_molecules, 2023, doi:10.3390/molecules28031139_

Round 1
Reviewer 1 Report
The manuscript highlights the recent findings regarding the role of Ach system in the common respiratory diseases with a special focus on the involvement of nAChR in Cholinergic Anti-inflammatory Pathway, mechanisms understudied in vivo.
The authors should explain α7nAChR in more detail since it is a main receptor in Cholinergic Anti-inflammatory Pathway. What about their expression in the airways and lungs? Are there any other nicotinic receptors relevant for Cholinergic Anti-inflammatory Pathway?
What about possible adverse effects of α7nAChR agonists use in COPD and Asthma, such as airway smooth muscle proliferation or possible interaction with anticholinergic drugs used in COPD treatment?
The tone authors use when speak about α7nAChR agonists use in respiratory diseases is rather optimistic, although there is a lot of controversy surrounding this matter. First of all, α7nAChR are nicotinic receptors and activated by nicotine, thus, smoking would not be detrimental for respiratory diseases, contrary to clinical data. Is there an explaination for this discrepancy? In a recent study, smoking was associated with increased α7nAChR expression and we know that smoking exarcebates asthma.
Are there any data relating interaction of α7nAChR with SARS-CoV-2 ?
Reviewer 2 Report
Congratulations to Authors on the very important, up to date, detailed, nice and interesting review.
Please find here some minor suggestions.:
Row 11: : "As an indispensable component in our human beings" the „our„ is not needed
: As an indispensable component in human beings
Row 18 multiple pathogens the word „ multiple” is not necessary here
Row 21: in this field of in the field of
Row 21: we still hope that we hope that
Row 18 : „Mycobacterium Tuberculosis” is typed in italics, „Severe Acute 18 Respiratory Syndrome Coronavirus 2” is not typed in italics, please choose whether you will apply or not apply italics
Row 33: „Acetyl- „ not needed capital letter A, enough the „a”: acetyl-
Figure 1: - next to the cross section figure of the bronchus there is a little mistyping: please add the letter „e” : …….and Mucus Hyper-Secretion (not „scretion”)
- the arrowheads of the square/angled/rectangular arrows (2 arrows) at the lower part of the figure 1, might be better to point to the post synaptic part (where M3/M1 and M3/M2 are shown)
Row 43: calcitonin gene-related peptide
Row 52: Optional: You can (if you would like to) insert some short statements about muscarinic receptors in the heart or other different organs. After it you can continue with the airways as you very nicely detailed.
Some suggestions only (not necessary).
You can consider citing Professor E. Sylvester Vizi’s article:
Vizi ES, Kobayashi O, Töröcsik A, Kinjo M, Nagashima H, Manabe N, Goldiner PL, Potter PE, Foldes FF. Heterogeneity of presynaptic muscarinic receptors involved in modulation of transmitter release. Neuroscience. 1989;31(1):259-67. doi: 10.1016/0306-4522(89)90048-1. PMID: 2549449.
https://pubmed.ncbi.nlm.nih.gov/2549449/
„Existence of three different subtypes of presynaptic muscarinic receptors is suggested: M1, which mediates increase of acetylcholine and noradrenaline release; M2 which mediates inhibition of acetylcholine release from the Auerbach plexus; and M3 which mediates inhibition of acetylcholine release from cerebral cortex and noradrenaline release from sympathetic axon terminals of the right atrium.”
OPTIONAL ONLY/NOT NECESSARY (Because this review focuses on the airway system) :
Other statements from Wang; Pappano:
The existence of M1, M2, M3, and M5 receptors in the human heart was shown with different methodologies [Wang et al.]. Reduction in ICa,L in heart cells is mediated by M2 receptors [Pappano].
Wang, H.; Han, H.; Zhang, L.; Shi, H.; Schram, G.; Nattel, S.; Wang, Z. Expression of multiple subtypes of muscarinic receptors and cellular distribution in the human heart. Mol. Pharmacol. 2001, 59, 1029–1036. [Google Scholar] [CrossRef] [PubMed]
Pappano, A.J. Cholinoceptor-activating & cholinesterase–inhibiting drugs. In Basic & Clinical Pharmacology, 13th ed.; Katzung, B.G., Trevor, A.J., Eds.; McGraw-Hill Education LLC: New York, NY, USA, 2015; pp. 105–120. ISBN 978-0-07-182505-4. [Google Scholar]
Rows 89 – 90 : „M2 and M3 mAChRs are expressed by mucus-producing cells, while M1 and M3 mAChRs can be detected in airway smooth muscle.”
Please check Figure 1 as at the bottom of the Figure 1 -depicting postsynaptic parts- (e.g. smooth muscle on Figure displays M3/M2) does not fully correlate with the text
(rows 89-90 in the text talks about M1 and M3 regarding smooth muscle)
Row 102: ACh Row 103: Ach Row 95: Ach Row 143: ACh
Row 163: Ach Row 168: ACh…. please check at all text whether ACh or Ach you use
Row 90: „M1 and M3 mAChRs can be detected in airway smooth muscle.”
-and later- in Row 105: „Moreover, M2 and M3 mAChRs can be found in airway smooth muscle with an M2 predominance.”
(please make homogenous in these parts)
Row 113: „ in (Table 1.)” Brackets are not needed
Row 131: „ in (Table 1.)” Brackets are not needed
Row 187 nucleus (not nuclear)
Figure 2: Bacterium (not bactrium)
Figure 2: Extracellular (not extracelluar)
Row 240: „anticholinergics are the recommended treatment for these diseases”
can be modified to : anticholinergics are in the treatment of these diseases
Row 260: mAChRs, expressed on the submucosal glands - should be altered to-
mAChRs are expressed on the submucosal glands
Row 267: provides significant clinical benefit in patients with COPD (than) ICS/LABA -„than” to be replaced :
provides significant clinical benefit in patients with COPD over ICS/LABA
Row 277: anti-inflammatory effect through their investigations - maybe a comma could be put in : anti-inflammatory effect, through their investigations
Row 298: found that selective α7nAChR agonist, -maybe you can use letter „a” -
found that a selective α7nAChR agonist,
Row 299: Alternaria Alternat (missing the „a” from its end) : Alternaria Alternata
Row 310: „ negatively corrected to „ please see: negatively correlated to
Row 315: J. Wu, et al J. Wu et al
Row 317: Recently, Ze-ping Hu et al demonstrated please check whether the Citation 95 contains the Researcher
Row 317: Please recompose the sentence below, because it might contain grammatical/composition errors e.g.: ACh-mediated : it is an adjective
or ACh mediated : it is a statement /
is needed?
(Maybe by splitting into 2 sentences would be better and then recomposing).
„Recently, Ze-ping Hu et al demonstrated that ACh-mediated drug tolerance partly through WNT signaling in an ACh M3R-dependent manner, constituting the ACh/M3R/WNT axis in epidermal growth factor receptor (EGFR)-mutant NSCLC and it may modulate persistent drug tolerance in EGFR-mutant lung cancer and impede tumor relapse through targeting acetylcholine signaling[95].”
In summary: please recompose this sentence for the better understanding.
Row 323: „Therefore, targeting muscarinic acetylcholine receptor-related preparations may become candidate drugs to exert anti-tumor effects shortly.”
„related” is not proper term here, alternatives: equipped/containing etc.
„shortly” is not a proper word here alternative: soon
Please recompose this sentence for the better understanding.
Row 332: point is not needed for the end: [100,101].
Row 331: New expression is needed, the „instead of” is not appropriate here:
„Instead of the fact that nicotine can induce the proliferation of endothelial cells[100,101]. it was found that nicotine also can induce the proliferation of a variety of small cell lung carcinoma cell”
Row 350: „Instead of” term is not proper, please use an other e.g.: Besides the α7nAChR, α5nAChR also
Row 374: …. targets and drugs in the treatment of this lethal malignancy …….
Row 380 and row 389 „M. tuberculosis” , „ Mtb „ used, please choose one of them, and please use that.
Row 378 after „Mycobacterium Tuberculosis” you can insert in brackets (Mtb) or sg. similar
Row 378 and 393 Mycobacterium Tuberculosis, Mycobacterium tuberculosis
Row 395: CC Chemokine ligand
Row 395 CXC
Row 405 and 406 : Not necessary this part: „In terms of the current, the COVID-19 epidemic is hard to be ended in a short time. However, „
You can start with: „Some studies…… „
Row 427: dose-dependent model or dose-dependent manner ? Please check
Row 430: Danial et al suggested……… At the references I did not find Danial et al at Reference No: (130)
Row 447: „They reported that nAChRs domains that share”
correct is : They reported that nAChRs domains share
Row 451: „contracting COVID-19” Please check the word: contracting
Row 460: „We strongly believe that there will always be a day when humanity will out- compete the SARS-CoV-2 virus.”
This sentence contains some grammatical errors and in my opinion this sentence is not needed here. It is obvious that we should eradicate the virus.
Row 476: „Even nowadays”, „Even” is not needed
Row 478: „ … cancer treatment. We hypothesize that………….. „ (please split to two sentences as shown)
Row 484: „developing targeted agents” targeted agents under development for the…
Reviewer 3 Report
DeHu Li et al., in this Review: The Role of the Acetylcholine System in Common Respiratory Diseases and COVID-19.
Its have discussed COVID-19 and the cholinergic system's anti-inflammatory pathway.
-Improve the figure resolution
- Standardizing acronyms and abbreviations: (E.g. Ach, ACh; M1, alpha-7)
